# Ultrabroadband high-resolution silicon RF-photonic beamformer

Pablo Martinez-Carrasco [1], Tan Huy Ho [2], David Wessel[2] & José Capmany [1] ✉

Microwave photonics aims to overcome the limitations of radiofrequency devices and systems by leveraging the unique properties of optics in terms of low loss and power consumption, broadband operation, immunity to interference and tunability. This enables versatile functions like beam steering, crucial in emerging applications such as the Internet of Things (IoT) and 5/6G networks. The main problem with current photonic beamforming architectures is that there is a tradeoff between resolution and bandwidth, which has not yet been solved. Here we propose and experimentally demonstrate a novel switched optical delay line beamformer architecture that is capable of achieving the desired maximum resolution (i.e., $2^M$ pointing angles for $M$-bit coding) and provides broadband operation simultaneously. The concept is demonstrated by means of a compact ($8 \times 3$ mm$^2$) 8 (5-bit) delay line Silicon Photonic chip implementation capable of addressing 32 pointing angles and offering 20 GHz bandwidth operation.

With the emergence of new communication paradigms like the Internet of Things (IoT) and 5/6G networks, the need for high-speed and high-capacity wireless communication has increased exponentially[1,2]. In this scenario, microwave-phased array antennas (PAAs) have become an indispensable element of contemporary radar and wireless communication systems[3]. Electronic phased array antennas operate by manipulating the phases of signals transmitted from each antenna element in the array. By doing so, the array can focus its radiation in a specific direction, resulting in high directivity and reduced interference from other directions[4]. However, this use of phase shifts can lead to the beam squint effect, where changes in the microwave frequency cause a deviation in the pointing angle, limiting the operating bandwidth to less than a few hundred megahertz[5–7].

Microwave photonics technology has enabled the introduction of optical true time delay line (OTTDL) technology into phased array PPAs[8] bringing key advantages in terms of low signal loss and immunity to electromagnetic interference[9] and more importantly, avoiding the appearance of the beam squint effect and extending the operating bandwidths of PAAs[10]. Numerous fiber-based architectures have been proposed for implementing OTTDLs, either based on Bragg gratings[11,12], highly dispersive fibers[13,14], or wavelength multiplexing[15,16]. While these approaches have yielded impressive

performance, they typically are bulky and require precise cleaving of their length to achieve accurate delays. On the other hand, on-chip integrated microwave photonics opens new possibilities with a reduction in space weight and power consumption (SWAP)[17–19]. In this area, there have also been multiple OTTDL proposals based on different architectures and technologies which can be grouped into two main categories. PAA architectures based on single and cascaded microring resonators (MRRs)[20–22], provide continuous tuning of the time delays, resulting in high angular resolution, but face bandwidth limitations arising from the resonant behavior of their delays. Furthermore, they need complex stabilization circuits to avoid resonance drifting due to temperature fluctuations, which are greatly increased when the number of antennas in the array escalates. PAAs based on switched M-bit delay lines can offer significant delays with wide bandwidth and low sensitivity to temperature fluctuations[23–25] at the expense of discrete tuning of the time delays. There is thus a tradeoff between resolution and bandwidth which has led to a recent proposal[26] to substantially increase the resolution of switched delay line PAAs by means of a parallel configuration of $N$ OTTDLs, where the basic delay increment is not constant in each OTTDL, instead it increases by one unit. This configuration is capable of achieving maximum resolution (i.e., $2^M$ pointing angles) but

[1]Photonics Research Labs, iTEAM Research Institute, Universitat Politècnica de València, Valencia, Spain. [2]Ottawa Wireless Advanced System Competency Centre, Huawei Technologies Canada Co., Ltd, Ottawa, ON, Canada. ✉e-mail: jcapmany@iteam.upv.es

surprisingly suffers from beam squint for negative pointing angles due to the impossibility of providing negative incremental delays. Unfortunately, this prevents broadband operation. A similar architecture based on MRRs has been recently proposed[27], but suffers from the same operational problems, contrary to what is considered a feature of using true time delay.

Here, we propose and experimentally demonstrate an architecture that is capable of achieving the desired resolution (i.e., $2^M$ pointing angles), providing broadband operation at the same time. This configuration is based on the inclusion of a delay equalization stage prior to the switched delay line array to avoid the need for negative incremental delays to implement negative pointing angles. We demonstrate the successful operation in a silicon photonics chip for a record spectral range spanning from 10 to 30 GHz (which is almost 3 orders of magnitude larger than the bandwidth reported for electronic beamformers) and provide the roadmap to enlarge this range to 50 GHz and beyond. Furthermore, we show via Monte Carlo simulation that this configuration is scalable and robust against design and fabrication deviations.

## Results

### Limitations of current switched delay line beamformers

Switched OTTDLs have been extensively studied and utilized in the literature due to their simple design and operation, which provides one of the most robust methods of obtaining optical delays. Figure 1a shows the simplest design of one of these delay lines, which in general comprises $M$ delay stages or "bits". Each bit unit is implemented by means of a tunable switch and two optical paths with different lengths: the short path is constant for all delay stages, while the longer path increases its length with the number of bits as $2^{M-1}$, $U$ where $U$ represents the path imbalance of the first stage. The tunable switch is

implemented by means of a balanced Mach–Zehnder Interferometer (MZI), composed of two 3-dB directional couplers and two parallel waveguides loaded with phase shifters as shown in the inset of Fig. 1a. A traditional beamformer based on switched OTTDLs is implemented by combining in parallel N OTTDLs identical to the one shown in Fig. 1b. Here, each of the identical OTTDLs feeds an independent radiating element. The number of addressable pointing angles is given by (see "Methods"):

$$N_a = 2 \left\lfloor \frac{2^M - 1}{N - 1} \right\rfloor + 1 \tag{1}$$

The parenthesis on the previous equation denotes the integer part of a real number. It is immediate to observe that this figure (closely related to the beamformer resolution) scales quite poorly with the number of delay stages and active elements, due to its inverse dependence on the number of antennas. Ideally, one would want the number of possible pointing angles to be independent of $N$ and as close as possible to $2^M$.

Zhu and co-workers[26] have recently proposed a novel architecture shown in Fig. 1c, which in essence keeps the parallel OTTDL configuration but where the value of the basic switched delay τ is changed by one unit in each adjacent OTTDL (U for line 1, 2U for line 2, 3U for line 3 and so on). This configuration brings, in principle, the advantage of enabling $2^M$ pointing angles as desired providing broadband (i.e., squint-free) operation. In[26] the authors reported a 5-bit $N = 8$ microwave photonic beamformer on silicon a silicon photonics chip and demonstrated the capability of addressing 32 angles from −75.51 to 75°.64 at 16 GHz (see Fig. 8c of ref. 27 and reproduction here in the center upper part of Fig. 1d). Although it is claimed that this

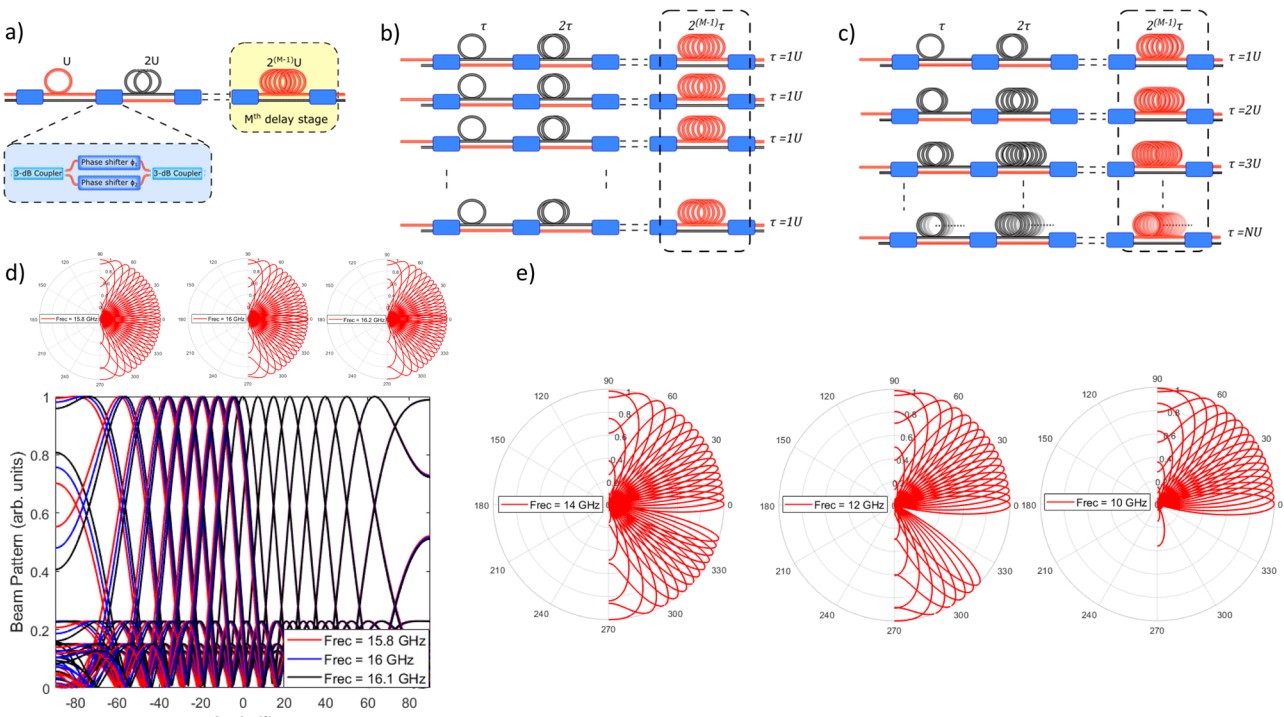

**Fig. 1 | Current beamforming network architectures based on OTTDLs. a** Scheme of switched OTTDL with $M$ delay stages. **b** Classical beamformer based on $N$ identical switched OTTDLs. **c** Beamformer composed of $N$ OTTDLs with increasing positive delays[26]. The unit delay value increases one time unit $U$ from one line to another (i.e., from $1U$ to $NU$). **d** Beam patterns generated by the architecture of ref. 26, for 15.8 GHz(red), 16 GHz (blue), and 16.1 GHz (black). In polar coordinates, a superposition effect becomes evident near the broadside region. Upon closer inspection of half of the total angles, the beam squint effect appears for negative pointing angles. **e** Beam patterns in the architecture of ref. 26, for 14, 12, and 10 GHz. In addition to the beam squint effect observed for negative angles the same angles begin to disappear for lower frequencies.

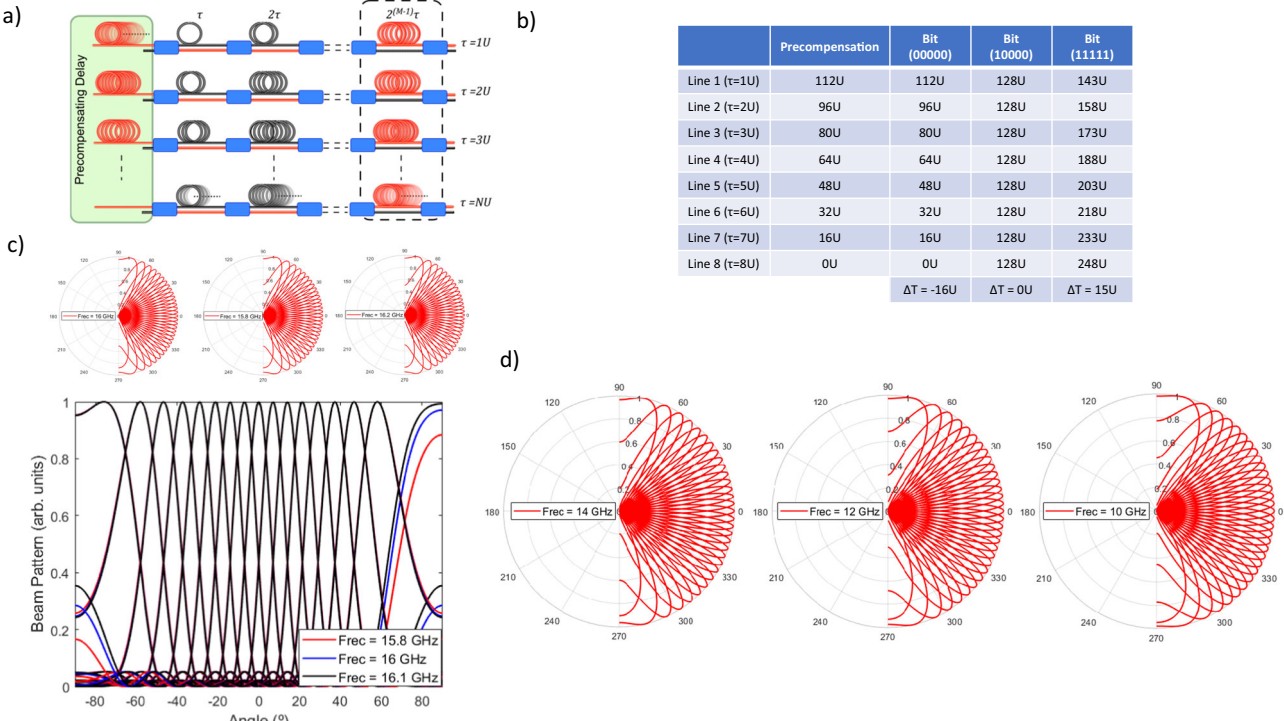

**Fig. 2 | Novel Beamforming network architecture with equalization. a** Layout of the architecture proposed in this work, emphasizing the pre-compensating/equalizing delay line stage component (highlighted in green). **b** Design table of the pre-compensation delay line stage for the unequalized 8 OTTDL beamformer and final overall output values for three different bit words, corresponding to the largest negative (00000), broadside (10000), and largest positive (11111) angle. **c** Beam patterns in the new architecture for 15.8 GHz (red), 16 GHz (blue), and 16.1 GHz (black). Beam squint for the negative angles has disappeared. **d** Beam patterns in the new architecture for 14, 12, and 10 GHz. All the pointing angles appear regardless of the operating frequency.

configuration is broadband (from 8 to 18 GHz) we found that beam squint is produced in the negative pointing angles even in a narrow frequency region (15.8–16.1 GHz) around this frequency as shown in Fig. 1d. Moreover, as the operating frequency is decreased to lower values (i.e., 14, 12, and 10 GHz), the number of negative pointing angles is reduced as well, as shown in Fig. 1e. This striking result is against the common belief that OTTDL-based beamformers should be beam squint-free but can be explained by an analysis of the Array factor of the beamformer (see "Methods" and Supplementary Note 1). The conclusion is that the scheme proposed in[26] is not able to provide broadband operation.

### Novel beamformer architecture

The main reason behind the beam squint is that the configuration reported in ref. 26 is not capable of providing negative incremental delays, which are essential to steer negative pointing angles independently of the frequency. These pointing angles, when achieved are due to phase shift and not true time delay conditions. A solution to overcome this limitation is to introduce a delay equalization stage prior to the switched delay line beamformer as shown in Fig. 2a. This equalization stage must be designed so the bit-word corresponding to the broadside radiation angle in the beamformer is rotated to the lowest negative radiation angle. In this way, both positive and negative pointing angles can be achieved using positive incremental time delays only, and therefore beam squint-free operation is achieved.

The equalization delay design aims to transform the incremental delay range of the beamformer from $[0, (2^{M-1})U]$ to $[-2^{M-1}U, (2^{M-1}-1)U]$. For line $n$, the value of the equalization delay must be therefore:

$$\epsilon_n = (N - n)2^{M-1}U \qquad (2)$$

The incremental delays between antennas are then:

$$\Delta T = T_n + \epsilon_n - T_{n-1} - \epsilon_{n-1} = BU - 2^{M-1}U \qquad (3)$$

Where $T_n$ and $\epsilon_n$ represent the selected delay and equalization delay for the line $n$. The value of the selected delay depends on the chosen bit-word $B$ (from $[0, (2^M-1)]$) and the position of the line as $T_n = nBU$.

And the array factor transforms to:

$$|F(\theta)| = \left[\frac{\sin\left(\pi f N\left[\frac{d}{c}\sin(\theta) - \left(B - 2^{M-1}\right)U\right]\right)}{N\sin\left(\pi f\left[\frac{d}{c}\sin(\theta) - \left(B - 2^{M-1}\right)U\right]\right)}\right] \qquad (4)$$

Where $d$ is the space between radiating elements in the array, $f$ is the RF frequency of the signal and $c$ is the speed of light.

From Eq. (4), we can compute the highest value for the basic unit delay that guarantees frequency-independent pointing angles. This is obtained by forcing the broadside condition ($B = 0$) in the unequalized beamformer and proceeding as shown in "Methods":

$$\frac{d}{c}\sin(\theta_{\min}) + 2^{M-1}U = 0 \qquad (5)$$

From Eq. (5), the maximum value allowed for $U$ given a negative minimum angle $-\theta_{\min}$ is $2^{1-M}(d/c)\sin(\theta_{\min})$ and it can be readily checked that for the maximum positive pointing angle ($B = 2^M - 1$):

$$\sin(\theta_{\max}) = \left[1 - \frac{1}{2^{M-1}}\right]\sin\theta_{\min} < 1 \rightarrow \theta_{\max} < \pi/2 \qquad (6)$$

Hence, all pointing angles fulfill the beam squint-free condition. Figure 2b provides the data for the design of the equalization stage for

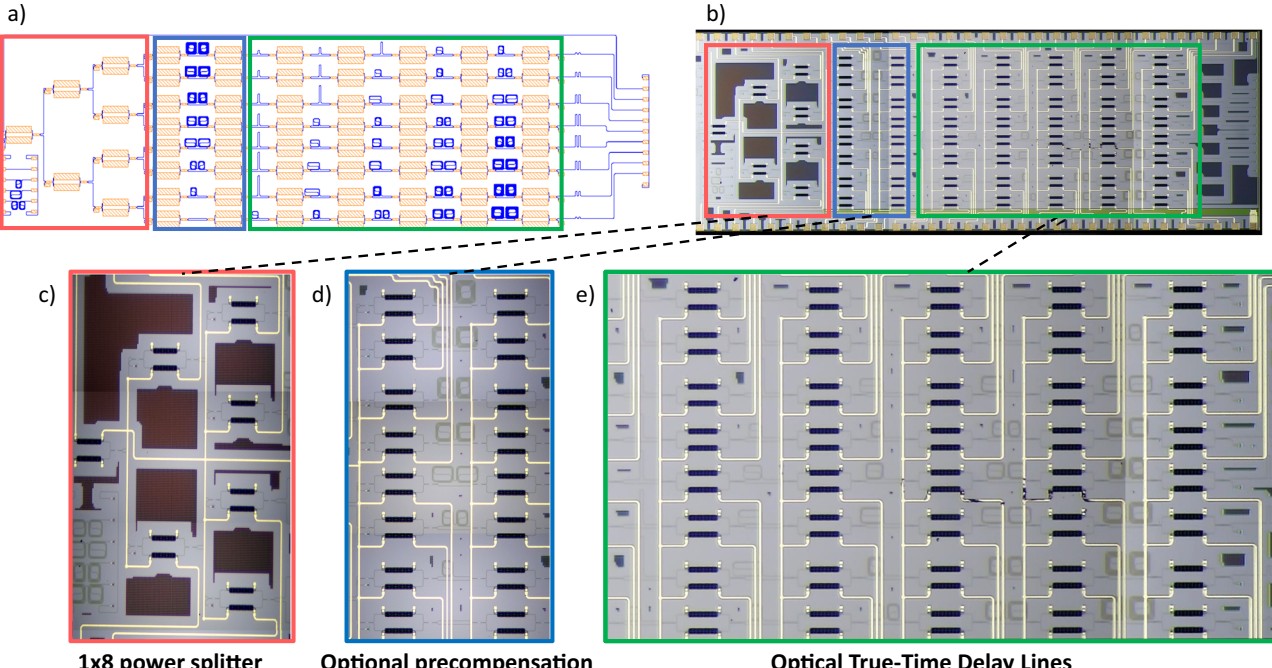

**Fig. 3 | Layout and micrographs of the silicon photonics beamformer chip.**
**a** GDS layout with the optical elements from the 1 × 8 equalized beamformer.
**b** Micrograph of the silicon photonics chip, including electrical connections
(8 mm × 3 mm). The three different parts are highlighted in color boxes (**c**) Zoom of
the 1 × 8 tunable splitter tree. **d** Zoom of the optional pre-compensating delay line
unit as an additional bit. **e** Zoom of the eight 5-bit OTTDLs stage.

the beamformer reported in[26] using Eq. (2) and the output delay results for each of the 8 OTTDLs in three relevant cases corresponding to the lowest negative pointing angle, broadside and the highest positive pointing angle (see Supplementary Note 2 for the complete table of values for the 32 pointing angles). Figure 2c shows the beam patterns for the same frequency range frequency region (15.8–16.1 GHz) as in Fig. 1d. Note that the beam squint has disappeared. Furthermore, Fig. 2d shows that as the operating frequency is decreased to lower values (i.e., 14, 12, and 10 GHz), the number of negative pointing angles does not change and their directions remain unaltered. These results confirm the simultaneous achievement of the maximum number of pointing angles $2^M$ and broadband squint-free operation.

### Beamformer chip design and fabrication
The beamforming network architecture has been demonstrated experimentally using a design with 8 OTTDL and 5 delay stages per line. For the demonstration, the equalization delay was added as an additional delay stage at the front end of each line. The design was made keeping in mind a PAA with a maximum operating RF frequency of $f_{max} = 30 GHz$ and a spacing between antennas of 5 mm, following the condition of single lobe of radiation $d \leq c/2f_{max}$. Based on the general design equation from Eq. (5), we calculated the value of basic unit delay setting a pointing angle range between −60° and 60°. For a value of $U = 0.92$ ps, we found the perfect balance between resolution and angle aperture while avoiding secondary lobes with too much intensity, resulting in an expected range from −61.87° to 55.7°.

The photonic integrated circuit (PIC) was manufactured in SOI technology by Advanced Micro Foundry (see "Methods"). Typical manufacturing figures from this foundry are: waveguide loss 1.1 dB/cm, MMI insertion losses 0.15 dB and grating coupler insertion losses 3.5 dB. The total insertion losses measured for the longest line configuration on the chip are 21.9 dB, taking into account 9 dB from the splitter tree and 7 dB from in/output grating couplers. All of these values for a wavelength of 1550 nm.

The phase shifter power consumption 1.35 mW/ π, with a tuning speed around 3 kHz. Low-power phase shifters implemented using suspended waveguide and trenches are key in enabling a reduced power consumption for the 65 MZIs integrated in the PIC, Fig. 3a depicts the chip layout, while Fig. 3b showcases a micrograph of the fabricated PIC (footprint measuring 8 mm × 3 mm). The three main sections of the beamformer: (a) The 1 × 8 tunable splitter, (b) the delay equalization stage and the 5 bit 8 OTTDL units (offering delays ranging from 0.92 ps to 117.76 ps) are shown in more detail in Fig. 3c–e, respectively.

### Experiments
The silicon photonic beamformer was tested and measured (see "Methods") using the setup depicted in Fig. 4a. A first characterization was performed to measure the phase and group delay frequency responses from 100 MHz to 30 GHz of each separate OTTDL unit for each of the 32 states that can be programmed using 5 bits. Figure 4b shows the results for the group delay characterization of OTTDL #1, #4, and #8 (see Supplementary Note 3 for the complete phase and group delay characterization of the 8 OTTDL in the beamformer). Here, the incremental delay is U = 0.92 ps for OTTDL #1, 4U = 3.68 ps for OTTDL #4 and 8U = 7.36 ps for OTTDL #8 as expected. Note that the group delay response shows an almost flat frequency response within the 30 GHz spectral range. We then proceeded to measure the frequency response of the delays provided by each of the 8 OTTDL units for each one of the possible 5-bit coding words and from them the beam patterns corresponding to each of the possible pointing angles. Figure 4c shows the simulated and measured beam patterns (at operation frequencies of 10, 20, and 30 GHz) for three pointing angles of the beamformer, corresponding to the lowest negative pointing angle, broadside and the highest positive pointing angle, respectively. Insets depict for each case the delays of each of the OTTDLs (see Supplementary Note 3 for the full measurement results). As it can be noted, the beam patterns are free from beam squint in a 20 GHz spectral range, which is over twice the highest value ever reported and furthermore, the number of available pointing angles is $2^5 = 32$, that is, the maximum possible for a 5-bit beamformer.

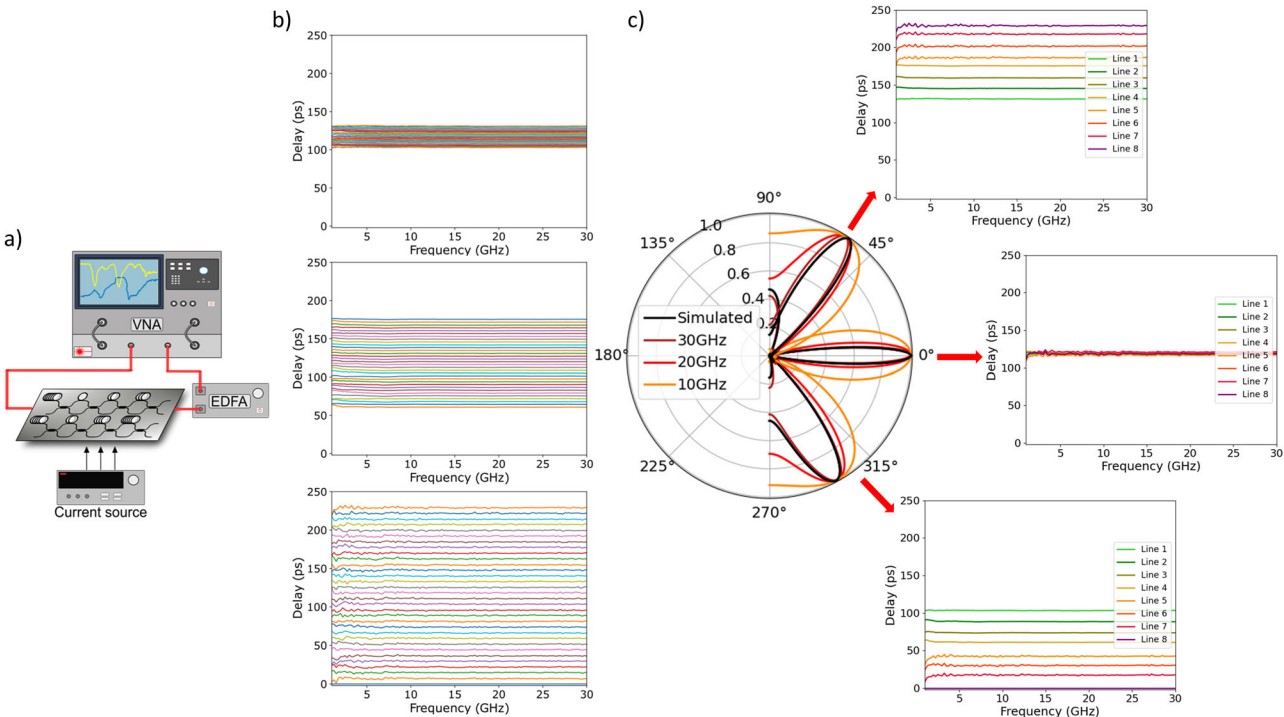

**Fig. 4 | Experimental setup and results. a** Experimental setup for the frequency response characterization of the Silicon photonics beamformer. **b** Group delay vs frequency measurement results for the first, fourth, and eighth lines (the results for the rest are included in Supplementary Note 3). Each one shows the 32 delay curves corresponding to the whole set of 5-bit configurations. **c** Calculated beam patterns vs three different frequencies (10, 20, and 30 GHz) for the largest negative (00000), broadside (10000), and largest positive (11111). Insets correspond to the group delay frequency responses of each pointing angle configuration (for full results of the 32 possible pointing angles, see Supplementary Note 4).

## Discussion

The proposed architecture provides a long-sought solution to the problem of achieving broadband operation and the highest possible number of pointing angles. It can be implemented using a compact layout, and there are several approaches to scale it in terms of OTTDL units and therefore of radiating antennas (i.e., from $1 \times 8$ to $1 \times 16$ or $1 \times 32$). A strategy to overcome the input power splitting loss in the beamformer is to use a Semiconductor Optical Amplifier (SOA) stage (preamplification configuration). InP SOA devices can routinely achieve gains in excess of 15 dB when incorporated into Silicon chips via micro-transfer printing. The number of bits per line can also be extended (i.e., from 5 to 7 or 8) leveraging the SOA gain but also by using novel low-loss building block designs in Silicon that can achieve propagation losses below 0.25 dB/cm with 2-μm waveguides[28,29], and reducing MZI insertion losses to 0.22 dB[30].

Another benefit of this architecture is its low-power consumption. For instance, for the design reported here, the average power consumption is 79 mW (see details in Supplementary Note 4).

There are, however, several limitations that need to be addressed. The first one is related to the expected power imbalance due to the different compensating lengths used for the OTTDLs. The propagation losses of each line depend on the number $n$ of the OTTDL and the pointing angle configuration chosen for the array:

$$L_n = e^{-\left(\frac{\alpha U}{20}\right)\frac{c}{n_g}Ln(10)\left[(N-n)2^{M-1}+nB\right]} = e^{-\gamma\left[(N-n)2^{M-1}+nB\right]} \quad (7)$$

where $\alpha$ is the propagation loss coefficient in terms of optical power of the technology and $n_g$ is the silicon waveguide group index. To facilitate the calculations, the constants have been grouped into a single constant, $\gamma$. This imbalance can act as a source of unwanted tapering, deviating the beam pattern from the desired configuration. To evaluate its impact, we can approximate the array factor by the field radiation pattern from a continuous apodized aperture of length $N$ as a function of an angle p:

$$p = f\left[\frac{d}{c}\sin(\theta) - \left(B - 2^{M-1}\right)U\right] \quad (8)$$

Which is given by:

$$|F(p,\gamma)|^2 = \left|\frac{1}{N}\int_{\frac{1}{2}}^{N+\frac{1}{2}} e^{-\gamma[(N-n)2^{M-1}+nB]}e^{-i2\pi px}dx\right|^2$$
$$= \frac{e^{-\gamma[B(N+1)+(N-1)2^{M-1}]}}{N^2\left[\left(\gamma\left(B-2^{M-1}\right)\right)^2+(2\pi p)^2\right]}\left[4\sin^2(N\pi p)-2+e^{-N\gamma[B-2^{M-1}]}+e^{-N\gamma[B-2^{M-1}]}\right]$$
$$(9)$$

Note that for an uniform (i.e $\gamma = 0$) field pattern, Eq. (9) gives the array factor of the uniform beamformer Eq. (4). The effect of imbalance loss can be evaluated though $\varepsilon(p,\gamma) = |(|F(p,\gamma)|^2 - |F(p,0)|^2)/|F(p,0)|^2|$, which for the beamformer reported here renders negligible results for waveguide propagation loss coefficients as high as $-3$dB/cm (even in this case it results in a negligible difference with respect to the uniform beam pattern in terms of shape, see Supplementary Note 5). In any case, full compensation and even intentional apodization can be implemented by inserting p-n junction-based variable optical attenuators (VOAs) at the output of each line or by the use of a variable splitter tree.

Unavoidable fabrication errors and the evaluation of their potential impact on the beamformer sensitivity is also a main concern. By identifying potential issues early on, we can avoid costly and time-consuming errors that would otherwise require significant resources to fix[31,32]. The impact of fabrication errors can be estimated using statistical techniques and simulation. Of particular importance is the

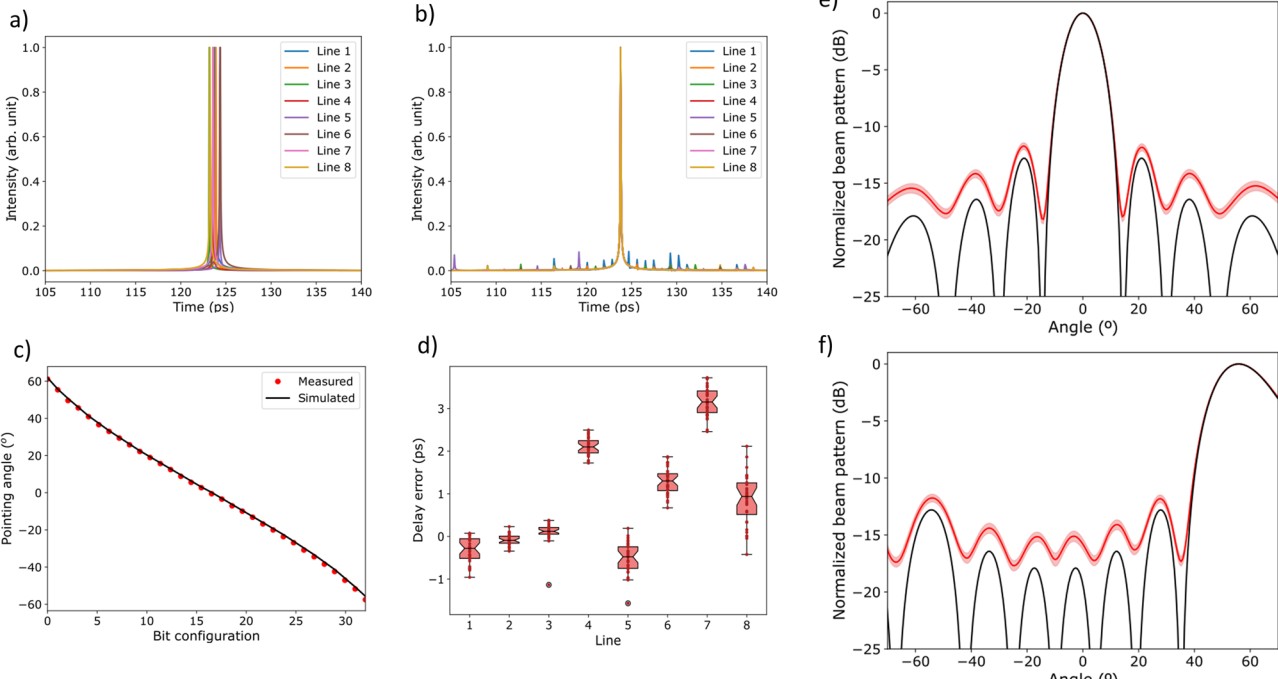

**Fig. 5 | Tolerance of the equalized beamformer to fabrication errors. a** Impact of fabrication errors in tunable couplers $\sigma_K$ and $\sigma_\phi$ over the broadside beam pattern (small amplitude random spike pulses). **b** Impact of deviations in the waveguide group index $\sigma_{ng}$ over the broadside beam pattern (deviations from the mean delay). **c** Experimental measurements of the impact of delay errors in our chip for all pointing angles (at 30 GHz). The boxes extend from the first quartile (Q1) to the third quartile (Q3) of the data, with a notch at the median. The whiskers extend from the box to the farthest data point lying within 1.5× the inter-quartile range (IQR) from the box. The median values of delay error from the first to the latest line are: −0.35 ps, −0.12 ps, 0.11 ps, 2.05 ps, −1.24 ps, 0.52 ps, 2.45 ps, and 0.1 ps. **d** Experimentally measured vs theoretically expected pointing angles in the reported silicon photonics beamformer at 30 GHz. **e, f** Results of a Montecarlo analysis over 1000 realizations of the impact of fabrication errors over the broadside (**e**) and highest positive pointing angle (**f**).

impact of common variations on the pointing angles, which arise from the asymmetric power splitting ratio on the 3-dB couplers, as well as from deviations in the phases of both arms of the MZI switches. Also relevant are the changes in the waveguide dimensions, which affect the group index in different sections of the structure[33,34]. For the evaluation, the best option is to use a Monte Carlo analysis. The operation of each delay stage and MZI switch from each OTTDL in the beamformer can be modeled by three Gaussian random variables centered at their ideal values, having a standard deviation, $\sigma_k$, $\sigma_\phi$, and $\sigma_{ng}$ for random fluctuations around the mean of the coupling coefficient (K), phase term ($\phi$) and group index, respectively. Manufacturing imperfections in parameters related to MZIs provided by different foundries and technologies are typically given by $\sigma_K = \sigma_\phi = 1\%$[35,36]. As far as the group index is concerned, this is highly sensitive to the specific technology used, the manufacturing technique, and any errors in waveguide dimensions. Fortunately, there are many available studies on these deviations for various integrated photonics technologies[37], providing information on manufacturing deviations from the usual group indices. For our experimental demonstration, the chip was fabricated at a standard 220 nm-thick Silicon on Insulator foundry. With these characteristics, knowing the type of guide and the manufacturing standards, we conclude that the expected group index is $n_g = 4.18$, with a variance of $\sigma_{ng} = 1.2\%$. Figure 5a, b shows an example of the impact (over one realization) of manufacturing errors on the time delays for the eight different outputs of the beamforming network in the case of broadside radiation. For this case, all the delays provided by the beamformer lines should be equal. Figure 5a. Illustrates de impact of $\sigma_K = \sigma_\phi$, which translate into small amplitude random spike pulses. The impact of $\sigma_{ng}$ shown in

Fig. 5b translates into slight random deviations from the mean delay. Experimental results on measured delay errors in our chip are shown in Fig. 5c. Despite variations of up to 3 ps in the worst cases, this error is well-tolerated when reconstructing the corresponding beam patterns, as shown in Fig. 5d.

To account for a more accurate description we ran a Monte Carlo process with 1000 realizations to evaluate the impact of errors on the beam patterns corresponding to the broadside and the highest positive pointing angles. The results are shown in Fig. 5e, f, respectively, and reveal that manufacturing errors impact performance mainly by reducing the Main Lobe to Secondary Ratio (MLSR) less than 1.5 dB, with negligible impact on the main lobe characteristics and their pointing angles, varying the beamwidth at 3 dB less than 1°. This indicates that the proposed beamforming architecture should provide excellent resilience to fabrication errors.

In conclusion, we have reported and experimentally demonstrated a novel tunable beamforming architecture based on silicon photonic integrated switched delay lines that is capable of providing simultaneous record broadband and squint-free operation and the maximum number of pointing angles $2^M$, where M is the number of coding bits. The key novel concept resides in inserting an equalizing stage to enable negative pointing angles using exclusively positive true time incremental delays. We have specifically demonstrated squint-free operation from 10 to 20 GHz of an 8-lines 5-bit OTTDL Beamformer, capable of addressing 32 pointing angles ranging from −61.87° to 55.7° with a remarkably low power consumption (below 100 mW). The features provided by this new architecture go remarkably beyond

**Table 1 | Performance comparison of various state-of-the-art integrated microwave photonic beamformers**

| Technical approach/ reference | Structure | Platform | Space (mm²) | Bandwidth (GHz) | Delay range (ps) | Power cons. (mW) | Ins. losses (dB) | Angle range (°) | Angles | Lines |
|---|---|---|---|---|---|---|---|---|---|---|
| Switched Delay Lines[26] | MZI-TTODLs | SOI | 11.03 × 3.88 | 10(*) | 0–496 | 1450 | 35 | 0 to 75.64 | 16 | 8 |
| Switched Delay Lines[38] | MZI-TTODLs | SiN | 8 × 32 | NS | 0–22.5 | NS | NS | −51 to 34 | 6 | 4 |
| Switched Delay Lines[24] | MZI-TTODLs | SOI | 7.4 × 1.8 | NS | 0–191.4 | 1251 | 27 | – | – | 1 |
| Switched Delay Lines[28] | MZI-TTODLs | SOI | 3.9 × 5.6 | NS | 0–210.6 | NS | 10 | −60 to 60 | 9 | 16 |
| Switched Delay Lines[29] | MZI-TTODLs | SOI | 3.8 × 10.9 | 23.5 | 0–310.8 | 4560 | 11-7 | NS | 9 | 8 |
| Continuous[39] | MRR | SiN | 2 × 11 | 5 | 34–252 | NS | 11 | −28 to 34 | – | 2 |
| Continuous[40] | MRR | SOI | 4.575 × 0.8 | 2 | 36–200 | 862 | NS | −30 to 30 | – | 4 |
| Continuous[41] | MRR | SiN | 8 × 32 | 8.6 | 0–208 | NS | NS | −22 to 22 | – | 4 |
| Switched Delay Lines (this work) | MZI-TTODLs | SOI | 8 × 3 | >20 | 0–228.2 | 72 | 12.9 | −55.7 to 61.87 | 32 | 8 |

(*) Only for positive beam steering angles.

the current state of the art as shown in Table 1 and furthermore this configuration can be readily extended to both broader spectral regions (i.e., 10–50 GHz operation can be readily achieve by reducing $U$ from 0.92 to 0.55 psec, which is possible with current Silicon photonics fabrication technologies) as well as to the higher number of radiating elements (from 8 to 16 or 32) and 2D beam steering by means of chiplet configurations. These flexible features make them a strong candidate to support massive MIMO architectures in future 6 G systems.

## Methods

### Array factor of a true time delay 1D beamformer

The performance of a 1D beamformer from the pointing angle characteristic is given by its Array Factor:

$$F(\theta) = \sum_{n=1}^{N} \frac{1}{N} e^{-i2\pi f\left[\frac{d}{c}\sin(\theta) - T_n\right]}$$

$\theta$ is the angle describing the pointing direction and $T_n$ is the time delay for antenna element $n$. We can define $\Delta T$ as the time delay between successive antenna elements given in general by:

$$\Delta T = T_n - T_{n-1} = nBU - (n-1)BU = BU$$

Where $U$ is the basic unit delay. For traditional beamformer architecture $B$ ranges from $-E[-(2^M-1)/(N-1)]$ to $E[(2^M-1)/(N-1)]$, where $E$ represents the integer part. For the resolution-improved architecture proposed here $B$ ranges from 0 to $2^M - 1$. By developing the summation and taking the absolute value we arrive at:

$$|F(\theta)| = \left[\frac{\sin\left(\pi f N\left[\frac{d}{c}\sin(\theta) - BU\right]\right)}{N\sin\left(\pi f\left[\frac{d}{c}\sin(\theta) - BU\right]\right)}\right]$$

The squint-free pointing angles of the beamformer are determined by:

$$\frac{d}{c}\sin(\theta) - BU = 0$$

Provided that $|cBU/d| \leq 1$. Note as well that if the condition is fulfilled and $B$ is positive then squint-free angles are contained in the $[0,\pi/2]$ region. If the condition is not fulfilled, then the pointing angles are given by:

$$\frac{d}{c}\sin(\theta) - BU = -\frac{k}{Nf}$$

With $k = 1,2,3...$ which are frequency dependent and thus lead to beam squint.

### Fabrication

The designed photonic integrated circuit was manufactured by Advanced Micro Foundry, according to a standard SOI process. The chip was fabricated from a SOI wafer with a 220 nm slab thickness, with 500 nm single-mode waveguides defined using deep ultraviolet lithography (193 nm). The waveguide sections with phase shifters are obtained by depositing a thin heater layer over the waveguide, made of 120 nm TiN, which will be powered by metal DC tracks of 2000-nm thickness deposited in the final manufacturing stages.

### Phase and group delay measurements

The phase and group delay were measured using a phase-shift approach where RF-to-RF response is measured after going through the integrated circuit. In the laboratory, the chip was integrated into a test and measurement setup, where input and output signals were injected and extracted using optical fibers connected to the corresponding grating couplers (one input coupler and eight output couplers). The switched OTTDLs were programmed using thermo-optic phase shifters, which were controlled by electrical currents from an external source array. The input RF-modulated signal was generated by a Vector Network Analyzer Agilent N5245A capable of modulating signals up to 50 GHz. With a lightwave component analyzer module N4373C, we upconverted the signals to the optical domain using a 1550-nm internal source. Afterward, the output signal from each OTTDL was also downconverted using the same optical module from the VNA. The measured group delay response is obtained after normalizing the measurement by taking into account the external optical components employed in the experimental setup of Fig. 4a.

## Data availability

All raw and processed data generated in this study are available from the corresponding author upon request.

## Code availability

All codes are available from the corresponding author upon request.

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

## Acknowledgements

The authors acknowledge the financial support of Huawei through project YBN2020065124, Microwave-Photonic IC Systemization and Development UPV. J.C. and P.M. acknowledge the support of the Valencian Government via the projects PROMETEO/2021/015, IDIFEDER/2018/031, IDIFEDER 2020/32, and IDIFEDER/2021/050.

## Author contributions

P.M. and J.C. carried out the original design and calculations for the beamformer, P.M. designed the silicon chip and carried out the measurements. P.M., J.C., and T.H. analyzed the results and J.C., T.H., and D.W. managed the project.

## Competing interests

The authors declare that a patent has been applied for the main concept reported in the paper.
