## [Peer Review File · Nature Communications]

Ultrabroadband high-resolution silicon RF-photonics beamformerEditorial Note: This manuscript has been previously reviewed at another journal that is not operating a transparent peer review scheme. This document only contains reviewer comments and rebuttal letters for versions considered at *Nature Communications*.

REVIEWER COMMENTS

Reviewer #1 (Remarks to the Author):

For the novelty, both two novelties mentioned by the authors have been proposed by their conference paper as mentioned previously. So it is not original.

For the Reply. 4, why do four of the delay lines have much smoother spectral response than the rest? If the ripples were introduced by the environmental and lab condition variations. It is better to redo the test. In the figure, the delay fluctuation fourth line (158.7 ps) is larger than 4 ps, this is much larger than their assumption in the discussion. Besides, it is incorrect to just use the average value of the delay over the operation bandwidth. At different frequencies, the authors should use the exact delay from the measurement to calculate the beam pattern

For the Reply. 5, does the normalization account the delay variation between different delay channels? Which state is the normalized 0-ps delay?

For the insertion loss of the chip, the authors just gave the assumed values according to the results provided by the foundry. It is important for the authors to give the real measurement results of all the delay lines to show the wavelength dependence and the fabrication uniformity. Besides, the figure in Reply.6 should be replaced by the real test results. For the S21, although theoretically the S21 response is flat in frequency, the fabrication deviation would induce some imperfection which will degrade the beam pattern. I disagree that the authors just give the assumed values for these important results for the chip.

For the Reply. 9, since the authors give no structure and analysis about the 2D beamformers, it is not appropriate to give the statement in the abstract. Besides the incremental loss for the 64×64 2D beamformer is also inconceivable.

Reviewers' Comments:

Reviewer #1 (Remarks to the Author):

For the novelty, both two novelties mentioned by the authors have been proposed by their conference paper as mentioned previously. So it is not original.

We thank the reviewer for pointing out this concern. Regarding submitted papers that have been previously published in conference proceedings, the editorial policy of Nature Group Journals is the following:

*The Nature journals are happy to consider submissions containing material that has been published in a conference proceedings paper. **However, the submission should provide a substantial extension of results, methodology, analysis, conclusions and/or implications over the conference proceedings paper**; the final decision on what constitutes a substantial extension is made by the editors at each individual journal. Authors must provide details of the conference proceedings paper with their submission including relevant citation in the submitted manuscript.*

In what follows we outline the reasons why we believe that our paper provides such a substantial extension both qualitatively and quantitatively. First of all the attached table provides a quantitative comparison of the extension, number of figures, tables and sections for each paper

	Conference Proceedings paper	This submission
Manuscript pages (only text)	3 (double column)	28 (Main) + 9 (Supplementary)=37
Number of simple figures ¹	23	37 (Main) + 86 (supplementary)=123
Number of tables	1	2 (Main) + 2 (supplementary) = 4
Number of independent sections	5	5(Main) + 4(Supplementary)

¹ We consider a simple figure any graphical item that can be considered independent from the others

Quantitatively there is thus an overwhelming increase in the material that is included in both submissions. This increase results from both enlarging the material of the conference proceeding paper as well as from the inclusion of novel material that is not included in that submission:

- **(Extension)** The complete derivation of the origin of beam-squint in the un-equalized true time delay beamformer architecture (enlarged from 1 paragraph to 1 page in the methods section and 2 pages in the Supplementary material)
- **(Extension)** The complete equalizer design tables for the 8 lines from 3 cases to all the possible 32 cases.
- **(Extension)** Chip design and fabrication, extended from 90 to 318 words.
- **(New)** the complete differential group delays between the 8 delay lines for each of the 32 programming cases from DC to 30 GHz, with the corresponding beam pattern computed at 10, 20 and 30 GHz.
- **(New)** Full analysis of the effect of power imbalance due to the different compensating lengths used for the OTTDLs.
- **(New)** Analysis of the impact of fabrication errors on the beamformer performance (528 words and 6 simple figures)
- **(New)** Analysis of the beamformer power consumption (152 words and one table)
- **(New)** A comparative table of recent published approaches for implementing integrated beamformers with the relevant performance parameters including those of our beamformer.

For the Reply. 4, why do four of the delay lines have much smoother spectral response than the rest? If the ripples were introduced by the environmental and lab condition variations. It is better to redo the test. In the figure, the delay fluctuation fourth line (158.7 ps) is larger than 4 ps, this is much larger than their assumption in the discussion. Besides, it is incorrect to just use the average value of the delay over the operation bandwidth. At different frequencies, the authors should use the exact delay from the measurement to calculate the beam pattern.

Reply:

Thank you for your comment. The reason for this behavior is that our measurements were conducted on different days, with four lines measured on one day and the remaining four on another with the calibration measurements executed only once. Hence, since the stability of various factors, such as the setup, fibers, RF cables, and vibration, can vary from one day to the next we believe these were the source for the observed differences.

To confirm this, we have repeated the measurements of all the lines using the same calibration conditions and included the new results in the revised manuscript. As expected, the ripples have disappeared.

With respect to the average value comment: All the reconstructed beam patterns, both in the main text and supplementary use the precise temporal measurement of each line at the desired frequency.

Actions:

We have remade the measurements for all the line configurations and computed the corresponding beampatterns. All the figures in the Main and Supplementary have been updated with the new measurements.

For the Reply. 5, does the normalization account the delay variation between different delay channels? Which state is the normalized 0-ps delay?

Reply:

Thank you very much for your comment. All lines are normalized to the smallest optical path traveled in the beamformer, i.e. the eighth line at a position where no delay is added (bit word 00000). This is therefore the configuration with the lowest delay, 0 ps, and is the one used to normalize all measurements.

Actions:

We have added the following sentence to Supp. Note 3

“All measurements were taken using as reference the smallest optical path within the beamforming network, the eighth line for bitword 00000.”

For the insertion loss of the chip, the authors just gave the assumed values according to the results provided by the foundry. It is important for the authors to give the real measurement results of all the delay lines to show the wavelength dependence and the fabrication uniformity. Besides, the figure in Reply.6 should be replaced by the real test results.

We thank the reviewer for the comment and apologize for the misunderstanding. The reported values are not those from the foundry but the result of real measurements. As the reviewer rightly points out, it is important to study the possible variability in the manufacturing processes and this is something of which our research group is very aware. Prior to the measurement of any complex design, large quantities of measurements are taken of the individual components in order to obtain a solid statistical basis of the system's losses and its wavelength behavior.

We have measured the spectrum for the extreme cases that are represented by the first and eighth lines in the array. In each case, we measured the maximum (bit word 11111 orange) and minimum (bit word 00000 blue) delays. Each measurement has been normalized to the response of the grating couplers and the maximum power measured on each line for the bit word 00000.

As it can be seen, the wavelength response once normalized to the response of the grating couplers is flat. Both graphs illustrate how losses increase due to propagation for larger bit configurations.

We would like to state that the chip has no optical packaging, which means that the input and output fibers must be manually aligned with the grating couplers. This manual alignment process gives a variation in the measured power much higher than the propagation losses due to the line configuration. We believe that the following image will help to clarify the problem.

As can be seen in the image, it is necessary to re-align the output port when changing lines, which makes it impossible to measure the absolute insertion losses, since most of the variation will be induced by ourselves.

For the S21, although theoretically the S21 response is flat in frequency, the fabrication deviation would induce some imperfection which will degrade the beam pattern. I disagree that the authors just give the assumed values for these important results for the chip.

Thank you very much for your comment, we understand the author's concern as any manufacturing defect can influence the electro-optical response of the device. Fortunately, the foundry we work with has great precision in its processes and has never given us any problems.

We have made measurements of the electro-optical S21 for the eighth line of the beamformer in order to demonstrate its flat frequency behavior. First of all, we have calibrated the measurement to the setup used, since the elements that will most influence the S21 are the modulator and the photodetector, both integrated in the VNA.

After that, we have measured the response when passing through the line in different configurations. Once we normalize the measurements, we obtain a very flat frequency response whose level variations are only due to propagation losses for the different configurations.

For the Reply. 9, since the authors give no structure and analysis about the 2D beamformers, it is not appropriate to give the statement in the abstract.

Besides the incremental loss for the 64×64 2D beamformer is also inconceivable.

We thank the reviewer for this comment. We would like to point out that we do not explicitly mention the existence of a 2D array, or do suggest that one is on the horizon. Our statement is an educated guess that this architecture is likely to see future use, mainly because it performs well, can scale effectively, and is very robust against thermal variations.

To avoid misunderstandings, we have deleted the statement from the abstract

REVIEWER COMMENTS

Reviewer #1 (Remarks to the Author):

Thanks to the authors for putting in a lot of effort to respond to comments and redo some experiments. But I still have some questions.

1. In the manuscript, the authors gave the loss of some key elements. However, each component has a wavelength-dependent loss. The authors should mention the loss value at which wavelength in the manuscript. Besides, it is better for the authors to put the measured spectra of these key elements in the supplementary materials.
2. In the discussion, the propagation loss in Eq. 7 is the optical loss. To calculate the array factor, it should transfer to the microwave loss. The statement "even in this case it results in a difference with respect to the uniform beam pattern of only 0.06%", is also ambiguities. The difference is the shape, or the SLSR?
3. Other than the loss imbalance, I think the large loss of the beamformer is also a big concern for its scalability. The authors can put the discussion in the manuscript.
4. In Fig. 5(c), is the delay errors at the frequency of 30 GHz? From supplementary Figure 5, it seems the delay errors are larger at the low frequencies. The authors should make it clear and it is better to explain why the delay errors are larger at the low frequencies.

Reviewers' Comments:

Reviewer #1 (Remarks to the Author):

1. In the manuscript, the authors gave the loss of some key elements. However, each component has a wavelength-dependent loss. The authors should mention the loss value at which wavelength in the manuscript. Besides, it is better for the authors to put the measured spectra of these key elements in the supplementary materials.

Thank you for your suggestions. We will indicate the working wavelength and we will add all the material to the supplementary.

Actions:

We have added the following sentence in the Main text:

“All of these values for a wavelength of 1550 nm.”

In addition, we have included the information regarding the measurements carried out for the discrete components of our chip in Supplementary Note 6, which is reproduced here in its entirety.

Supplementary Note 6

Before measuring the beamforming network, we collected data to estimate the performance of the fundamental building blocks employed in the final design, in order to characterize their insertion losses and wavelength dependence.

All the structures show a flat response in wavelength and the fittings were done for the principal wavelength of use, 1550 nm.

Propagation Losses measurements

We used the cutback technique to measure the propagation losses in straight waveguides of different lengths incorporated as test structures in the chip as shown in Supplementary Fig. 8

Supplementary Figure 8. Structure used to measure the propagation losses of the waveguides. Input by grating coupler array.

The main results for the spectral dependence of losses in the 1520-1580 nm wavelength range are shown in the left part of Supplementary Fig. 9, where each color corresponds to a waveguide of different lengths. From those, we can extrapolate for a given wavelength a linear regression curve as shown in the right part of Supplementary Fig. 9 for 1550 nm. At this wavelength the slope corresponds to a loss of 1.1 ± 0.3 dB/cm.

Supplementary Figure 9. Wavelength spectrum and linear fitting for the different lengths. The responses correspond to each of the different lines, each one longer than the previous one. The propagation loss coefficient obtained is 1.1 ± 0.3 dB/cm.

MMI Losses measurements

Similarly, we used a set of test MMI configurations as shown in Supplementary Fig 10 to allow us to measure the insertion losses of a cascade of MMIs ranging from 1 to 10.

Supplementary Figure 10. Structure used to measure the insertion losses of the MMIs. Input by grating coupler array.

The main results for the spectral dependence of losses in the 1520-1580 nm wavelength range are shown in the left part of Supplementary Fig. 10, where each color corresponds to a cascade of different numbers of MMI couplers. From there, we can extrapolate for a given wavelength a linear regression curve as shown in the right part of Supplementary Fig. 10 for 1550 nm. At this particular wavelength, the slope corresponds to an insertion loss of 0.15 ± 0.05 dB/MMI.

Supplementary Figure 11. Wavelength spectrum and linear fitting for a different number of concatenated MMIs, from 1 to 10. The slope analysis provided us with the insertion loss value, obtained after accounting for the power loss resulting from the MMI's 50/50 splitting ratio. Our measurement indicates an insertion loss of 0.15 ± 0.05 dB/MMI.

MZI measurements

Finally, we characterized the insertion losses for different switching MZI units employed to setup the signal routing in each delay line according to the programmed bit word. The main insertion losses versus the wavelength results are shown in Supplementary Fig. 12, where the different curves correspond to different currents applied to the thermo-optic actuators (0 to 0.7 mA). The extreme values correspond to bar and cross states while the intermediate correspond to partial coupling. As in the previous cases, the results at 1550 nm were employed to find the MZI insertion loss at that wavelength (in the bar state) yielding a value of 0.45 dB.

Supplementary Figure 12. The colored lines show the wavelength response for different currents applied to the thermo-optic actuators. On the right side, the response of one of the MZI at 1550 nm.

2. In the discussion, the propagation loss in Eq. 7 is the optical loss. To calculate the array factor, it should transfer to the microwave loss. The statement "even in this case it results in a difference with respect to the uniform beam pattern of only 0.06%", is also ambiguities. The difference is the shape, or the SLSR?

Thank you very much for your comment. We try to explain it better in the main text.

Our idea behind this theoretical study was to prove that even though we have different propagation lengths and losses in the eight delay lines, these do not create any relevant undesired apodization to the final beampattern.

We performed the study taking into account only propagation losses, as this aspect often raises questions regarding the architecture employed. Of course, there would be a reduction in directivity and intensity in the microwave signal due to these losses, but the shape of the beampattern is still that of a uniformly fed array, which is what we wanted to highlight with the sentence: "even in this case it results in a difference with respect to the uniform beam pattern of only 0.06%"

Actions:

We have added/revised the following sentence in the main document:

"Even in this case it results in a negligible difference with respect to the uniform beam pattern in terms of shape, see Supplementary Note 5."

And we have added the following information and figures as extra material in the Supplementary Note 5.

Supplementary Note 5

Effect of loss imbalance in the Array Factor

Since the delay line stages in the beamformer have different waveguide lengths one may ask whether these can yield significant differences in propagation losses, which might potentially induce an undesired apodization effect in the beam pattern. Here we report the main simulation results of our architecture for different waveguide loss coefficients, ranging from 0 to 5 dB/cm (including our measured value of 1.1 dB/cm). Losses > 3 dB/cm are far beyond the worst-case scenario on any foundry nowadays, but we have nevertheless included them to prove that the apodization effect is negligible.

Supplementary Figure 6. (a-d) Losses per line for 0.75, 1.5, 2.25 and 3 dB/cm as loss coefficient. **(e-h)** The corresponding side angle (bitword 00000).

Supp. Figure 6 (a-d) illustrates the effect of various waveguide propagation loss values in each of the delay lines across all the 32 5-bit word configurations. Due to the precompensation stage, the most substantial dissimilarities for the extreme side angles, whereas for the broadside

configuration, all lines exhibit uniform propagation losses. Therefore, in Supp. Fig. 6 (e-h) we plot the corresponding beam patterns for one of the two extreme side angles (-61°). Inevitably, as the propagation losses increase the directivity and the sidelobes' extinction ratio decreases. However, in terms of its shape, the pattern closely retains its ideal form compared to that obtained with a uniform apodization, even in the presence of losses. In Supp. Fig. 7 we show the variation of the Main Lobe to Secondary Ratio (MLSR) with the loss coefficient for the worst-case extreme side pointing angle, proving that the value of tapering is negligible.

Supplementary Figure 7. The value of the MLSR for different propagation losses. In order to observe an appreciable change, we have to allow for propagation losses much higher than those routinely attainable in the current state of the art.

3. Other than the loss imbalance, I think the large loss of the beamformer is also a big concern for its scalability. The authors can put the discussion in the manuscript.

We would like to express our gratitude to the reviewer for this insightful comment. Undoubtedly, losses represent a significant challenge for anyone working in the field of integrated photonics, especially when considering scalability and mass production. Fortunately, we are witnessing continuous advancements in this domain, leading to notable reductions in propagation and insertion losses.

Significantly, in recent developments related to silicon on insulator technology, we've observed two separate studies that achieved remarkable propagation losses below 0.25 dB/cm, primarily through the utilization of 2 μm width waveguides [29,30].

Furthermore, ongoing research has successfully lowered insertion losses in MZIs to as low as 0.22 dB [31]. These developments open the door for companies like *Lightelligence* to manufacture products based on MZI matrix multipliers with an impressive order of 64x64.

Taking into account that the packaging technique has also improved significantly, achieving couplings losses of only 1.7 dB/facet [30], the result is that the most limiting element in terms of losses is not photonic, but the number of antennas used and the necessary power splitter.

The figures display graphs that compare the losses on each line when scaling the number of delay stages and the number of beamformer lines. It is evident that as the number of delays increases, the losses remain relatively stable. However, when the number of lines is increased, the power splitters cause losses to escalate rapidly.

Actions

We have rewritten/added information on the following paragraph from the Discussion.

“The proposed architecture provides a long-sought solution to the problem of achieving broadband operation and the highest possible number of pointing angles. It can be implemented using a compact layout and is scalable in terms of OTTDL units and therefore of radiating antennas (i.e. from 1x8 to 1x16 or 1x32) as well as in the number of bits (i.e. from 5 to 7 or 8). Furthermore, it is low power consumption. For instance, for the design reported here, the average power consumption is 79 mW (see details in Supplementary Note 4).”

New paragraph:

“The proposed architecture provides a long-sought solution to the problem of achieving broadband operation and the highest possible number of pointing angles. It can be implemented using a compact layout and there are several approaches to scale it in terms of OTTDL units and therefore of radiating antennas (i.e. from 1x8 to 1x16 or 1x32). A strategy to overcome the input power splitting loss in the beamformer is to use a Semiconductor Optical Amplifier (SOA) stage (preamplification configuration). InP SOA devices can routinely achieve gains in excess of 15 dB when incorporated into Silicon chips via micro-transfer printing. The number of bits per line can also be extended (i.e. from 5 to 7 or 8) leveraging the SOA gain but also by using novel low-loss building block designs in Silicon that can achieve propagation losses below 0.25 dB/cm with 2 μ m waveguides [29,30], and reducing MZI insertion losses to 0.22 dB [31].

Another benefit of this architecture is its low power consumption. For instance, for the design reported here, the average power consumption is 79 mW (see details in Supplementary Note 4).”

[29] Xie Y., Hong S., Wu J. et al. Low-Loss Wavelength-Selected Tunable Optical Delay Lines for Microwave Photonic Signal Processing. *IEEE International Topical Meeting on Microwave Photonics (MWP)* (2023).

[30] Ni Z., Lu L., Liu Y., et al, Silicon-Integrated 8-Channel 6-bit Tunable Optical True-Time Delay Lines with High Switching Speed and Low Loss. *IEEE International Topical Meeting on Microwave Photonics (MWP)* (2023).

[31] Pai, S., Sun, Z., Hughes, T. W., Park, T., Bartlett, B., Williamson, I. A., ... & Miller, D. A. (2023). Experimentally realized in situ backpropagation for deep learning in photonic neural networks. *Science*, 380(6643), 398-404.

4. In Fig. 5(c), is the delay error at the frequency of 30 GHz? From supplementary Figure 5, it seems the delay errors are larger at the low frequencies. The authors should make it clear and it is better to explain why the delay errors are larger at the low frequencies.

We thank the reviewer for pointing out this concern. Yes, the errors have been calculated for the maximum operating frequency, which is 30 GHz, as indicated in the label in the figure.

Regarding the behavior at low frequencies, this is due to the loss of precision in our RF Vector Network Analyzer, especially below 1 GHz. In any case, we are talking about a really small deviation and a range of frequencies from DC to 2 GHz max, which is far below the limit of use of the beamforming network, intended to be from 10 GHz to 30 GHz. When it is used at lower frequencies the beamwidth grows rapidly, decreasing the directivity of the beamformer.

This beamwidth increase depends on the following theoretical expression for the beamwidth at 3 dB (N number of antennas, c speed of light, f frequency, d space between antennas).

$$\theta_{3dB} = \sin^{-1} \left(\sin(\theta) + \frac{0.433c}{Ndf} \right) - \sin^{-1} \left(\sin(\theta) - \frac{0.433c}{Ndf} \right)$$

REVIEWERS' COMMENTS

Reviewer #1 (Remarks to the Author):

The authors have put great efforts to show more experimental results. I have one more question. In table 1, the author should also compare the insertion loss of these delay line chips as well. Besides, Refs. [29] and [30] should be included in table 1 to represent the latest work.

RESPONSE TO REVIEWERS:

Reviewer #1 (Remarks to the Author):...I have one more question. In table 1, the author should also compare the insertion loss of these delay line chips as well. Besides, Refs. [29] and [30] should be included in table 1 to represent the latest work.

We have incorporated the missing information in Table 1. Regarding the data and code availability we have left a very open statement that declares that both are available upon request. We have not left the data in any public database as they are mainly results from measurements using particular instruments available in the lab and the code is a standard MATLAB routine to process the data. Upon request we will be happy to share details about both.